# Effect of Titanium and Boron Microalloying on Sulfide Stress Cracking in C110 Casing Steel

**DOI:** 10.3390/ma13245713

**Published:** 2020-12-15

**Authors:** Ming Luo, Zhong-Hua Zhang, Yao-Heng Liu, Mou-Cheng Li

**Affiliations:** 1Institute of Materials, School of Materials Science and Engineering, Shanghai University, 149 Yanchang Road, Shanghai 200072, China; luo_ming@baosteel.com; 2Baosteel Research Institute, Baoshan Iron & Steel Co., Ltd., Shanghai 201900, China; zhzhang@baosteel.com (Z.-H.Z.); liuyaoheng@baosteel.com (Y.-H.L.)

**Keywords:** casing steel, Ti and B, sulfide stress cracking, prior austenite grain size, hardenability

## Abstract

The effect of Ti and B microalloying on the hardenability, prior austenite grain size (PAGS), mechanical properties, and sulfide stress cracking (SSC) of C110 grade steel was studied by Jominy testing, static tensile testing, an optical microscope (OM), scanning electron microscopy (SEM) and double cantilever beam (DCB) testing. The results show that the addition of 0.015% Ti and 0.002% B into a medium-carbon Fe-Cr-Mo-Nb-V steel increased the hardenability and refined the PAGS and quenched martensite packets, and the size of carbides was reduced. It is believed that these behaviors are responsible for the improvement in the threshold stress intensity factor K_ISSC_.

## 1. Introduction

Stress-corrosion cracking (SCC) in a wet hydrogen-sulfide environment poses the greatest threat in oil and gas fields. The current mining depth of oil and gas in western China has reached 8000 m. Casing and tubing steels in oil and gas fields in high-pressure and high-corrosion environments require a high strength and a high corrosion resistance. The American Petroleum Institute (API) added C110 grade casing and tubing steel in the latest API Spec 5CT standard. During the production of oil country tubular goods, the steel’s chemical composition and heat treatment are the main factors that can be optimized to obtain the most suitable microstructure with a high strength and superior sulfide stress cracking (SSC) resistance. Steels with ferrite + bainite (F+B) and ferrite + martensite/austenite (F+M/A) dual-phase microstructures have a better hydrogen-induced cracking resistance, whereas the harder martensite phase in the F+M microstructure was responsible for the worst hydrogen-induced cracking resistance [1].

In the production of high-strength oil pipes with a yield strength of 728–827 MPa or more, according to API Spec 5CT, the phase ratio after quenching must contain at least 95% martensite. Therefore, the design of a high-strength sulfur-resistant tube should be based on Cr-Mo-V steel with further microalloying additions to ensure high hardenability, uniform mechanical properties over the wall thickness and high corrosion resistance in H_2_S environments [2,3].

The addition of a small amount of boron (several tens of ppm) increases the hardenability of low-carbon low-alloy steels and replaces certain expensive alloy elements, such as Cr and Mo [4]. The optimum content of boron to maximize the hardening effect in steel is in the range of 10 to 30 ppm. Outside this range, the hardening effect of boron disappears [5]. The effectiveness of boron depends on the addition of strong nitride formers, such as Ti, to prevent boron nitride formation [6].

The steel strength is enhanced gradually with an increase in the content of alloy strengthening elements, which is attributed to a combination of precipitation strengthening and prior austenite grain refinement [7]. The research and development of high-strength anti-sulfur tubes is difficult. As the steel strength increases, the sulfide stress corrosion decreases [2]. The threshold stress intensity factor K_ISSC_ of steels for SSC has a linear relationship with the lath size and martensite hardness; a smaller lath size and lower hardness result in higher critical stress intensity factor (K_ISSC_) values [8,9]. The precipitated phases and the ratio of the high-angle grain boundaries (HAGBs) influence the sulfur resistance of C110 steel. The increase in volume fraction of the nano-scale precipitated phase and HAGBs may enhance the SSC resistance [10,11]. The martensite structure that forms during high-temperature tempering treatment is composed mainly of body-centered cubic ferrite matrix with carbide particles, which favors the steel’s SSC resistance [12]. In the past, it was generally accepted that duplex stainless steel (DSS) was quite resistant to SCC in aqueous environments containing H_2_S, CO_2_ and chlorides [13]. However, the high content of Cr and Ni in DSS increases the cost, which restricts its application in oil-related tubular goods.

Previous studies rarely investigated SSC susceptibility by combing DCB testing with the Jominy test. In the present work, DCB tests performed on API Spec 5CT C110 casing steel specimens with the same quenching and different tempering temperatures are correlated with the specimens’ quantitative SSC resistance. The effect of Ti and B on anti-sulfur properties of C110 steel with the Fe-Cr-Mo-Nb-V steel was studied. The purpose was to study the mechanism of Ti and B in steel and to develop high-performance casing steel with a high strength and high SSC resistance.

## 2. Experimental Procedure

Each test steel with chemical composition in Table 1 was prepared by melting in a vacuum furnace, casting into a round ingot, heating and forging into square bars, and hot-rolling into a 15 mm-thick plate. TB0 and TB1 represent steels without and with the addition of 0.015% Ti and 0.002% B, respectively.

The hardenability of forged samples was tested according to the ISO 642 standard. The hardness was measured by a Rockwell hardness tester (Shanghai Optical Instrument Factory, Shanghai, China) with a load of 1500 N; each HRC hardness was measured three times and averaged for Jominy test specimens. To obtain a high-temperature tempered martensite, which is a microstructure with a desirable combination of strength and SSC resistance, the following quenching and tempering (Q+T) processes were performed on the test steels. TB0 and TB1 were austenitized at 920 °C for 60 min followed by water cooling. They were tempered for 2 h at 700 °C for TB0 and at 710 °C for TB1. All specimens were air cooled to room temperature after tempering. After polishing, quenched samples were eroded in 60 °C saturated picric acid, and the PAGS was measured by optical microscopy Axio Imager A2m (Zeiss, Jena, Germany) using the following equation [14]:(1)$$S=\frac{\pi d^{2}n}{4}$$
where *d* is the diameter of prior austenite grains, *S* is the total area of all grains in the analyzed optical microscopy images, and *n* is the number of the prior austenite grains.

After grinding and polishing, the sections of two specimens were etched with dipping absorbent cotton in 4% nitrate alcohol, and the precipitated phases were observed by secondary electrons on scanning electron microscope EVO MA25 (Zeiss, Jena, Germany) equipped with a tungsten filament gun. The size distribution of precipitates and segregations of alloying elements of the specimens were further studied using transmission electron microscopy JEM-2010 operating at 200 kV in scanning transmission electron microscopy (STEM) mode using the high angle annular dark field (HAADF) detector equipped with an energy dispersive X-ray spectroscopy (EDXS) unit. The test coupons were machined in parallel to the rolling direction to a gauge section of the standard tensile test sample to evaluate the mechanical properties. The static tensile tests were conducted using an MTS C40 (MTS, Eden Prairie, MN, USA) electronic universal testing machine with a 38.2 mm-gage length extensometer at ambient temperature with a strain rate of 2 × 10^−4^ s^−1^. The yield strength and tensile strength were measured according to ISO 6892-1 and API Spec 5CT. The equilibrium phase diagrams of the tested steels were calculated by Thermo-calc software.

The SSC susceptibility of the tempered specimens was evaluated by using several wedge-loaded DCB samples [15]. A double taper wedge with a suitable thickness was selected to provide arm displacement and loading by insertion into the DCB samples. Tests were conducted in an environment with a circulating solution that consisted of 5.0 wt.% NaCl and 0.5 wt.% CH_3_COOH that was saturated with 1 atm H_2_S. The solution was kept at a constant hydrogen-ion concentration and pH 2.7 at ambient temperature. Tests were performed for 336 h. According to NACE TM0177, the K_ISSC_ equation was found experimentally and modified to:(2)$$K_{\mathrm{ISSC}}=\frac{Pa(2\sqrt{3}+2.38h/a){\left(B/B_{n}\right)}^{1/\sqrt{3}}}{Bh^{3/2}}$$
where *P* is the equilibrium wedge load that was measured in the loading plane; *a* is the final crack length, which is quantified at the exposed crack surfaces by opening the DCB test specimen mechanically after wedge removal; *h* is the height of each arm; and *B* and *B_n_* are the specimen and web thickness, respectively. Three samples in each heat-treatment process were subjected to parallel DCB experiments under the same circumstances and conditions. The average K_ISSC_ values of the samples were expected to represent their SSC resistance.

## 3. Results

### 3.1. Microstructure Characterization

The optical morphologies of prior austenite grains after water quenching at 920 °C are shown in Figure 1. The PAGS is visible in TB1 steel with Ti/B elements compared with TB0 steel without elements Ti and B. The average grain diameter was ~6.1 μm for TB1 and ~7.8 μm for TB0. The addition of Ti and B reduces the PAGS and average grain diameter of steel.

Figure 2 shows the SEM microstructures of two steels. A martensite structure was produced in the specimens after quenching, and different direction martensite packets with almost parallel laths existed in the prior austenite grain. The average packet sizes that were measured from five pictures were ~4.5 μm for TB0 and ~3.3 μm for TB1. These results indicate that the addition of Ti and B results in a decrease in martensite packet size in steel.

The SEM micrographs of two steels after Q+T are shown in Figure 3. Tempered martensite was produced in the two steels, which consisted of recrystallized ferrite grains and spheroidized carbides (cementites). The carbides precipitated along grain boundaries and within the grains. The average carbides size of TB0 steel is ~80 nm, which is larger than that of TB1 steel (~60 nm). More large carbides existed along grain boundaries in TB0 steel compared with TB1 steel, which indicates that Ti and B addition may inhibit carbide growth. TB1 steel has a more uniform and dispersive carbide structure compared with TB0 steel.

### 3.2. Steel Hardenability

The quenching hardness curves are shown in Figure 4. To some extent, two steels with Ti and B and without Ti/B show a different hardenability. The hardness began to decrease from quenching surface at ~11 mm for TB0, and at ~17 mm for TB1, respectively. According to the requirements of the API Spec 5CT, the quenching microstructure must contain more than 95% martensite, so the minimum quenching hardness of C110 grade steel is [16]:(3)$${\mathrm{HRC}}_{\min}=59\times (\mathrm{wt}.\%C)+29=59\times 0.27+29\approx 45$$

According to the requirements of the minimum quenching hardness of API Spec 5CT, the single-sided depths for quenching exceeded 12 mm for TB0 and 18 mm for TB1, respectively. The addition of Ti and B improved the steel hardenability, which promoted the formation of uniform martensite in the entire wall thickness under the same quenching conditions.

### 3.3. Mechanical Properties and K_ISSC_ Values for Steels

Boron is one of the most effective elements to increase steel hardenability and strength. Steel strength is a key factor that affects the SSC performance. To eliminate the effect of precipitation strengthening from titanium, the candidate steels need to be quenched and tempered to a similar level. Figure 5a shows the room temperature tensile stress–strain curves of TB0 and TB1 steels. The mechanical properties and K_ISSC_ value of two steels are shown in Figure 5b. Both steels display similar average yield strength of ~780 MPa, the elongation-to-failure of the two samples reach the value of ~25%, and the average K_ISSC_ values are ~34.7 MPa·m^0.5^ for TB1 and ~27.5 MPa·m^0.5^ for TB0. The mechanical properties and K_ISSC_ values meet the requirements of the API Spec 5CT specification, which indicates that the yield strength is between 758 and 827 MPa and the K_ISSC_ value exceeds 26.3 MPa·m^0.5^. The yield strength of TB1 steel after tempering at 710 °C is similar to TB0 steel after tempering at 700 °C, but the K_ISSC_ value of TB1 steel is higher than that of the TB0 steel and the required minimum value of 26.3 MPa·m^0.5^.

## 4. Discussion

The corrosion products of 110S steel under the H_2_S and CO_2_ test conditions consisted mainly of different types of iron sulfides, and the absence of iron carbonate in the corrosion scales indicated that the corrosion process was controlled by H_2_S [17]. According to the description of the NACE MR0175 standard, low-alloy steels suffer from anodic dissolution, mainly as uniform and pitting corrosion. Hydrogen evolution occurs as the cathodic reaction. The hydrogen-evolution reaction must form hydrogen atoms by absorbing electrons from the anodic reaction. A small amount of hydrogen atoms form hydrogen molecules and evolve on the steel surface, but most hydrogen atoms penetrate the steel substrate. These hydrogen atoms gather at the weak spots, such as the grain boundaries, inclusions, segregations and dislocations, which can decrease the plasticity and toughness of steel and various cracking types occurs, such as SCC [18].

The tempered martensite of low-alloy steels has a large amount of carbide precipitates within the grains and in grain boundaries. Under external stress conditions, stress concentration occurs at the precipitated phases, and the stress concentration of the coarse precipitated phase is larger than that of the small and dispersive precipitation phase [19,20]. The coarser carbides that are precipitated in the matrix or along the grain boundary reduce the material’s toughness [21]. The high stress is assumed to produce a high dislocations density, whereas dislocations are strong hydrogen traps. Hydrogen can gather at the dislocation and form Cottrell atmospheres, which are transported by dislocations and accumulate in some areas. Because the hydrogen concentration exceeds a certain critical value, cracks will be initiated and propagate. Coarse carbides are more likely to cause a stress concentration, which may lead to the formation of a high surrounding hydrogen content, which promotes crack initiation through hydrogen embrittlement [22]. To obtain more information about carbide precipitation, the phase compositions of TB0 at 700 °C and TB1 at 710 °C are given in Table 2 by the calculation of equilibrium phase diagrams. The carbides include cementite, Fe and Mo precipitates (M_2_C), Fe and Cr precipitates (M_23_C_6_), V and Mo precipitates (MC) and Nb and Ti precipitates (Nb,Ti)C. The types of carbide that precipitate in the two steels are equivalent, but the large precipitations of M_2_C and M_23_C_6_ in TB1 are lower than that in TB0, whereas the small precipitates (Nb,Ti)C and MC in TB1 are higher than those in TB0. The calculation of the equilibrium phase diagram agrees with the SEM observation in Figure 3. Cr precipitates were found in Cr-containing steel with (Nb,Ti)(C,N) or Nb(C,N) as the precipitate core and then grew [23]. The precipitates of Cr-Mo steel were composed mainly of MC, M_2_C and M_23_C_6_ carbides. A sequence for the corresponding carbides transformation during tempering with the initial precipitation of MC and the subsequent precipitation of M_2_C and M_23_C_6_ was proposed [24]. Ti and B addition increased the nucleation sites of precipitation and prevented Cr and Mo carbide growth. Small carbide precipitation can improve the resistance to hydrogen sulfide stress corrosion [25].

The threshold stress *σ*_c_ of hydrogen induced cracking is [26]:(4)$$\sigma_{c}=3\left[\ln c_{th}-\ln(c_{H}-c_{t})\right]RT/\alpha V_{H}$$
where $c_{H}$ is the total hydrogen content in steel, $c_{t}$ is the hydrogen content in irreversible traps, $c_{th}$ is the critical hydrogen content to initiate cracking, *R* is the gas constant, *T* is the ambient temperature, α is the stress concentration factor of hydrogen-induced crack nucleation under uniaxial tensile stress, and *V_H_* is the partial molar volume of hydrogen in steel. With Ti/B addition into the steel, uniform and dispersive carbides formed in the steel, which may act as irreversible hydrogen traps to increase the $c_{t}$ value. The average sizes of the prior austenite grains and quenched martensite packets were reduced from ~7.8 to 6.1 μm and from 4.5 to 3.2 μm, respectively, with the refinement of tempered microstructure, which may increase the $c_{th}$ value. Equation (4) shows that higher $c_{t}$ and $c_{th}$ values result in larger *σ*_c_ value, which improves the SSC resistance of steel.

Because Ti/B microalloying exhibits a precipitation-strengthening effect, TB1 must be tempered at a 10 °C higher temperature to obtain a similar strength compared with TB0. The higher tempering temperature facilitates a reduction in steel dislocation density, which improves the SSC resistance [10,27].

Figure 6 shows the HAADF STEM images and EDXS mappings of the TB0 and TB1 specimens. Some Cr- and Mn-containing precipitates exhibited a bright contrast. The finer precipitates that contained Mo, Ti, and V formed in the TB1 specimen (Figure 6b). Except for Cr and Mn, enrichments of elemental Mo, Ti, and V were visible in the TB1, which indicates that (Nb,Ti)C and MC-type carbides (TiC,VC and (Ti,V)C) may precipitate during tempering. More precipitates exist in the TB1 specimen compared with the TB0 specimen, which proves that more small precipitates (Nb,Ti)C and MC exist in TB1 than those in TB0 (Table 2). Ti is a strong carbide-forming element in the steels, and it is much more competitive to react with C during casting to form TiC particles by increasing the nucleation sites of precipitation. According to grain size statistics in Figure 3c,d, the carbide size of TB0 steel is ~80 nm, which is larger than that of TB1 steel (~60 nm). The carbides with a size of ~70 nm or larger will lose their ability to trap hydrogen because of the incoherent interface between carbides and the matrix [28]. The nanosized MC carbides that act as irreversible traps retained the diffused hydrogen at room temperature [29,30,31]. The irreversible traps may mitigate hydrogen embrittlement by trapping the diffusible hydrogen and reducing the opportunities for hydrogen accumulation at potential crack nucleation sites [32]. Generally, the high-temperature-tempered martensite contains acicular ferrite and spheroidized carbides, the addition of Ti and B has the ability to suppress the coarsening of Cr and Mo carbides. Furthermore, the TB1 steel with finer carbides size possesses more semi-coherent interfaces, and thereby irreversibly trapping more hydrogen atoms [33]. As a result, the smaller carbide precipitations in TB1 steel can improve the resistance to hydrogen-sulfide stress corrosion.

## 5. Conclusions

The C110 steel (Fe-Cr-Mo-Nb-V) shows different quenched and tempered structures and K_ISSC_ values depending on 0.015% Ti and 0.002% B addition. It can be concluded that:Ti and B addition increases the hardenability of the medium-carbon Fe-Cr-Mo-Nb-V steel, and the single surface quenching depth of TB1 is higher by ~6 mm than TB0. Ti and B addition enhances the quenching hardness and tempering precipitation, which leads to the same strength level for TB1 that was tempered at 710 °C and TB0 tempered at 700 °C.Ti and B addition refined the prior austenite grains and martensitic packets after quenching and the microstructure and precipitated carbides after tempering, which improved the sulfide stress cracking of steel. The K_ISSC_ value of TB1 (34.7 MPa·m^0.5^) was approximately one quarter that of TB0 (27.5 MPa·m^0.5^).

## Figures and Tables

**Figure 1 materials-13-05713-f001:**
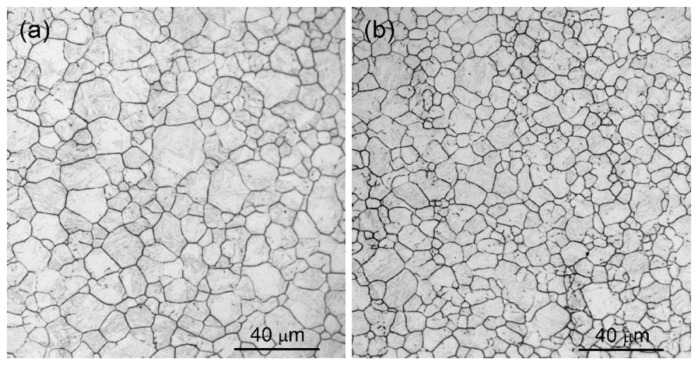
Optical morphologies of prior austenite grains of (**a**) TB0 and (**b**) TB1 steels after quenching.

**Figure 2 materials-13-05713-f002:**
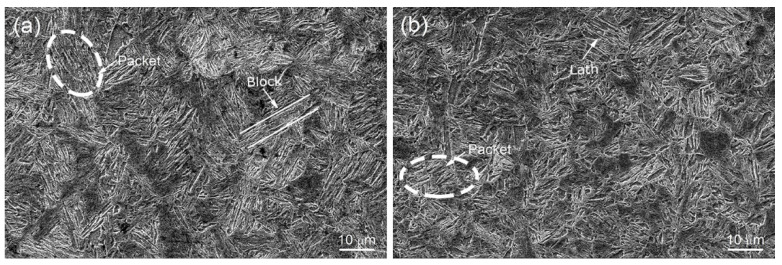
SEM micrographs show the martensite packets, blocks and laths of (**a**) TB0 and (**b**) TB1 steels after quenching and tempering (Q+T).

**Figure 3 materials-13-05713-f003:**
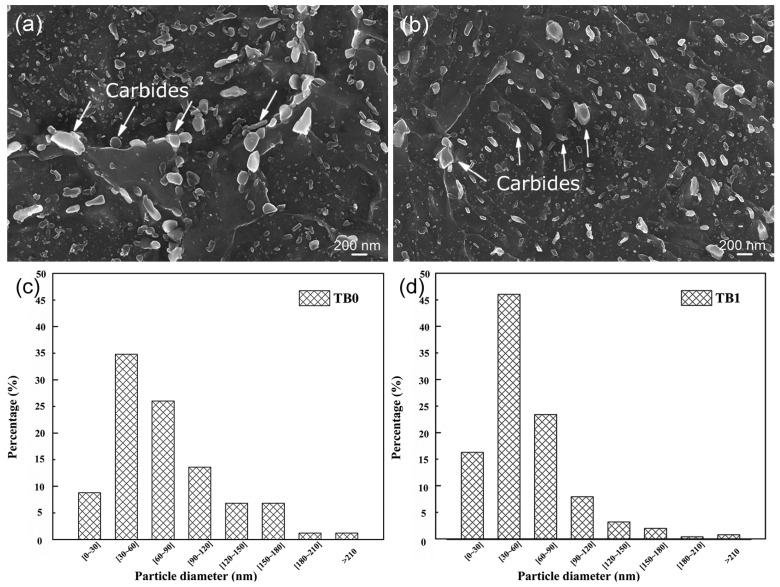
SEM micrographs of (**a**) TB0 and (**b**) TB1 steels after Q+T, and the corresponding carbide size distributions in (**c**) TB0 and (**d**) TB1 steels, respectively.

**Figure 4 materials-13-05713-f004:**
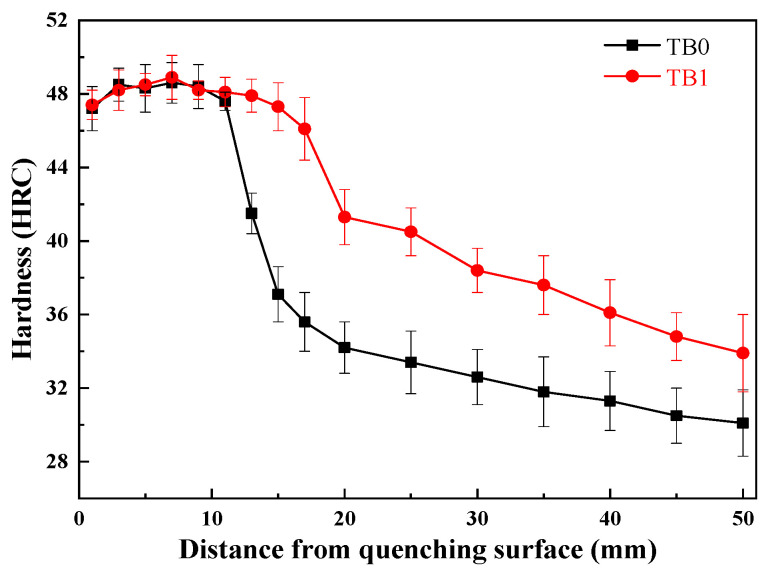
Quenching hardness curves of TB0 and TB1 steels.

**Figure 5 materials-13-05713-f005:**
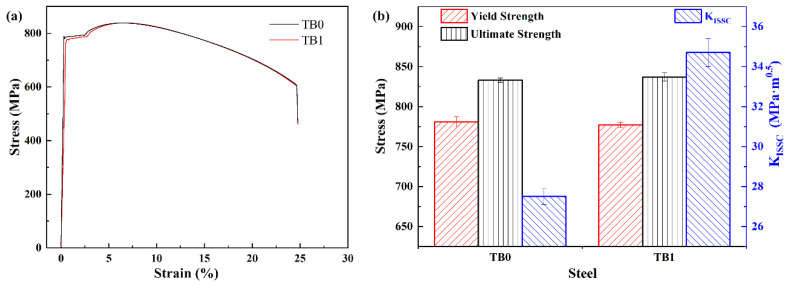
(**a**) Tensile stress-strain curves of TB0 and TB1 steels showing the similar mechanical properties, and (**b**) the values of yield and ultimate strengths and K_ISSC_ of TB0 and TB1 steels.

**Figure 6 materials-13-05713-f006:**
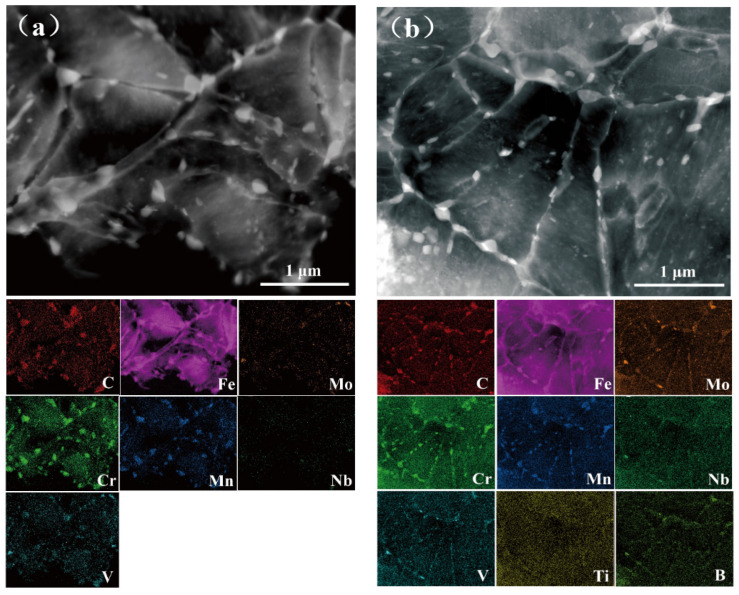
HAADF STEM images and corresponding EDXS mappings of (**a**) TB0 and (**b**) TB1 steels.

**Table 1 materials-13-05713-t001:** Chemical composition of the investigated materials (wt.%).

Steel	C	Si	Mn	S	P	Cr	Mo	V	Ti	Nb	B	N	O
TB0	0.27	0.28	0.49	0.001	0.009	0.51	0.87	0.10	—	0.04	—	0.004	0.001
TB1	0.27	0.28	0.48	0.001	0.009	0.53	0.78	0.11	0.015	0.04	0.002	0.004	0.001

**Table 2 materials-13-05713-t002:** Equilibrium phase content of steels after tempering (mol).

Steel	Ferrite	Cementite	(Nb,Ti)C	M_2_C	M_23_C_6_	MC
TB0	0.9560	0.0223	0.0007	0.0133	0.0037	0.0040
TB1	0.9568	0.0258	0.0010	0.0098	0.0022	0.0044

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
