# Peer review of "Effect of Titanium and Boron Microalloying on Sulfide Stress Cracking in C110 Casing Steel"

_materials, 2020, doi:10.3390/ma13245713_

Round 1
Reviewer 1 Report
Dear authors,
Please find attached file with my comments and suggestions.
Best regards

Author Response
The authors thank the reviewer for the detailed evaluation of their manuscript, especially the modification of sentences, words and grammars one by one. All of the reviewer’s suggestions have been incorporated into the revised version. The changes and modifications in the revised manuscript were highlighted using the “Track Changes” function in Microsoft Word to make it easily visible to the editors and reviewers.

Reviewer 2 Report
Referee Report
on paper “
Effect of Titanium and Boron Microalloying on Sulfide Stress Cracking in C110 Casing Steel”
by authors Ming LUO, Mou-Cheng LI, Zhong-Hua ZHANG, Yao-Heng LIU
submitted to Materials
This article examines the effect of adding Ti and B on the microstructure and mechanical properties of the casing (Cr–Mo–Nb–V) steel that are used in piping. The authors are trying to solve the urgent problem associated with cracking of the material under stress. It should be noted that the experiment was qualitative and good discussion, which confirms the high qualifications of the authors. The research results are presented clearly and logically ordered. As disadvantages, it is necessary to note imperfect English and the lack of a deep understanding of the mechanisms of interaction of additives with steel. The authors state the experimental results, but do not build a coherent line of reasoning. Nevertheless, the value of the article is high, thanks to the solution of the actual problem. In general, I highly appreciate the quality of the article despite insufficient discussion of the nature of the discovered effects and insufficient depth of analysis. I leave a number of comments below.
Comment 1. I recommend authors check English with a native speaker because a lot of sentences are hard to read or are written with syntax errors
Comment 2. Please explain the reasons for choosing the concentration of B (0.002 wt%) and titanium (0.015 wt %) in steel? Is it based on a previous experiment, theoretical background, or otherwise?
Сomment 3. Why were different tempering temperatures of steels chosen (700 °C for TB0 and 710 °C for TB1)?
Comment 4. The experimental part is incomplete. It is usually recommended to provide enough information to be able to repeat the synthesis process and all investigations after reading the article. For example, for scanning electron microscopy, the type of detector (secondary or backscattered electrons) and the accelerating voltage should be specified.
Comment 5. Information should also be given on how to estimate grain size. Was any software used and what was considered grain size? As a rule, the grain size is considered to be the size of a disk with an equivalent area, as in works “Structure of CrON coatings formed in vacuum arc plasma fluxes” 10.1016/j.surfcoat.2016.10.061 or “The Effect of Heat Treatment on the Microstructure and Mechanical Properties of 2D Nanostructured Au/NiFe System” doi:10.3390/nano10061077. This option seems to me the most acceptable in this case. How was this measured in this article?
Comment 6. I think that the article lacks an explanation of the nature of the discovered phenomena. The authors have shown that the addition of Ti and B from the steel improves the microstructure and results in acceptable cracking performance. Many authors very closely associate the mechanisms of cracking and other mechanical properties with microstructure. Please, see:
“Mechanical properties of Mo(C)N deposited using cathodic arc evaporation” doi: 10.1016/j.surfcoat.2017.04.005
“Mechanical properties of Cr-O-N coatings deposited by cathodic arc evaporation” DOI 10.1016/j.vacuum.2018.07.017
Here, it is not clear what caused such a significant change in the microstructure? What is the mechanism of interaction of additives with steel? This requires further discussion and is my main comment.
Author Response
The authors thank the reviewer for the detailed evaluation of their manuscript. All of the reviewer’s suggestions have been incorporated into the revised version. The changes and modifications in the revised manuscript were highlighted using the “Track Changes” function in Microsoft Word to make it easily visible to the editors and reviewers.

Reviewer 3 Report
The present work is devoted to the study of the effect of micro alloying on the stress cracking of C11 steel. The work is interesting however some major modifications are required before publishing in Materials in order to complete the work. Below you can find my specific comments.
1.Add and discuss in the introduction some works regarding possible alternative materials for the requested applications (for example DSS ” “K. van Gelder, J.G. Erlings, J.W.M. Damen, A. Visser, The stress corrosion cracking of duplex stainless steel in H2S/CO2/Cl− environments, Corrosion Science, Volume 27, Issues 10–11, 1987,Pages 1271-1279”)
2.State clearly at the end of the introduction section the novelty of the work in comparison with literature
3.I suggest to add in the figure caption of the micrographs the etch employed
4.Please add in Fig.3 proper labels in order to identify the different microstructural features (tempered martensite and carbides)
5.In Fig.3 and Fig.4 are reported several mechanical properties. Please add in the experimental section all the description regarding how these properties were evaluated
6.Regarding the properties of Fig.4 I suggest to add the tensile curves of the two steels to increase clarity and also considering that the paper is quite short, in order to understand better the differences in the mechanical behaviour of the steels
7.Also eventual differences in the impact toughness should be analysed considering that, as also stated by the authors, carbides influence a lot this property.
8.The composition of the carbides should be verified through X-ray diffraction with TEM or at least a tentative with normal XRD analysis should be performed (considering the quantity of carbides these could be seen also with XRD)
9.Whereas is clear the influence of the alloying on the mechanical properties the effect on the corrosion properties is not well discussed. I suggest to discuss more clearly this point in particular regarding the mechanism that induce the variation of KISSC in the two steels. Also a scheme that describe this mechanism will help for a visual understanding of the differences in term of corrosion mechanism
- Add and discuss in the discussion section some relevant work on the effect of carbides precipitation on the mechanical properties of structural steels (“Pezzato, L.; Gennari, C.; Chukin, D.; Toldo, M.; Sella, F.; Toniolo, M.; Zambon, A.; Brunelli, K.; Dabalà, M. Study of the Effect of Multiple Tempering on the Impact Toughness of Forged S690 Structural Steel. Metals2020, 10, 507” “Li, Z.; Jia, P.; Liu, Y.; Qi, H. Carbide Precipitation, Dissolution, and Coarsening in G18CrMo2–6 Steel. Metals 2019, 9, 916)
Author Response

(The authors gave the same response as above.)

Round 2
Reviewer 1 Report
Dear authors,
Please find attached file with my suggestions and questions.
Thank you
Best regards

Author Response
We are grateful to the reviewer again for revising the paper carefully. All suggestions have been incorporated into the revised manuscript, which were highlighted using the “Track Changes” function in Microsoft Word.

Reviewer 3 Report
Considering that the authors have answered to all the main issues of the first revision and have improved significantly the quality of the paper i suggest publication for this work after minor check of the english language.
